

**The Mid Atlantic Appalachian Orogen Traverse: A Comparison of Virtual and On-**
**Location Field-Based Capstone Experiences**
Steven Whitmeyer[1], Lynn Fichter[1], Anita Marshall[2], Hannah Liddle[1]
[1]Department of Geology and Environmental Science, James Madison University,
Harrisonburg, VA, 22807
[2]Department of Geological Sciences, University of Florida, Gainesville, FL, 32611-2120
Corresponding author email: whitmesj@jmu.edu





**Abstract**
The Stratigraphy, Structure, Tectonics (SST) course at James Madison University
incorporates a capstone project that traverses the Mid Atlantic region of the
Appalachian Orogen and includes several all-day field trips. In the Fall 2020 semester,
the SST field trips transitioned to a virtual format, due to restrictions from the COVID
pandemic. The virtual field trip projects were developed in web-based Google Earth,
along with other supplemental PowerPoint and PDF files. In order to evaluate the
effectiveness of the virtual field experiences in comparison with traditional on-location
field trips, an online survey was sent to SST students that took the course virtually in
Fall 2020 and to students that took the course in-person in previous years. Instructors
and students alike recognized that some aspects of on-location field learning were not
possible or effective with virtual field experiences. However, students recognized the
value of virtual field experiences for reviewing and revisiting outcrops, as well as noting
the improved access to virtual outcrops for students with disabilities, and the generally
more inclusive experience of virtual field trips. Students highlighted the potential
benefits for hybrid field experiences that incorporate both on-location outcrop
investigations and virtual field trips, which is the preferred model for SST field
experiences in Fall 2021 and into the future.
**1. Introduction**
On-location field trips and field experiences are a traditional component of
undergraduate geoscience curricula. However, the onset of the COVID-19 pandemic in
early 2020 resulted in quarantine restrictions that inhibited on-location fieldwork and
field-based educational experiences for at least a year. This left many geoscience
departments scrambling to find alternative field experiences for courses that traditionally
incorporated field-oriented educational components (e.g. Bond and Cawood, 2021;
Bosch, 2021; Gregory et al., 2021; Quigley, 2021; Rotzien et al., 2021.) The James
Madison University (JMU) Department of Geology and Environmental Science was
significantly impacted by pandemic-based field restrictions, as their traditional summer
capstone field course had to be reconfigured in a virtual format. Similarly, instructors for
several courses in Fall 2020 had to rethink how to conduct the field components of their
respective curricula. Among these courses was an upper-level geoscience course that
focuses on stratigraphic and structural analyses in the context of regional tectonics.
The JMU Stratigraphy, Structure, Tectonics (SST) course incorporates basic
principles of stratigraphy and basin analysis along with methods of structural analysis,
within the framework of models of the regional tectonic history and the Wilson Cycle
(Wilson, 1966; Burke and Dewey, 1974.) The course culminates with a multi-week
capstone project, where students spend 5 days in the field collecting stratigraphic and
structural data, while interpreting this data in the context of the Appalachian Orogen in
the Mid Atlantic region of western Virginia and eastern West Virginia (Fichter et al.,




2010; Figure 1.) This area is a classic example of relatively thin-skinned, fold and thrust
belt tectonics (e.g. Evans, 1989.) Most of the visible, outcrop-scale deformation in the
region resulted from the Alleghanian Orogeny (Bartholomew and Whitaker, 2010;
Whitmeyer et al., 2015,) although the Blue Ridge geologic province preserves
deformation and fabrics that derived from the Grenville orogenic cycle, as well as
younger Neo-Acadian high strain zones (Bailey et al., 2006; Southworth et al., 2010.) In
contrast, stratigraphic data from the field trips provide evidence for earlier tectonic
events, such as the Ordovician Taconic Orogeny and the Devonian Acadian Orogeny.
Students use stratigraphic and structural field data that they collect on the field trips to
draft a series of interpretive cross sections across the Blue Ridge and Valley and Ridge
geologic provinces, and then synthesize their data and interpretations in a report that
describes the tectonic history of the region, from the Mesoproterozoic Grenville orogeny
through the Paleozoic assembly of Pangaea (Whitmeyer and Fichter, 2019).

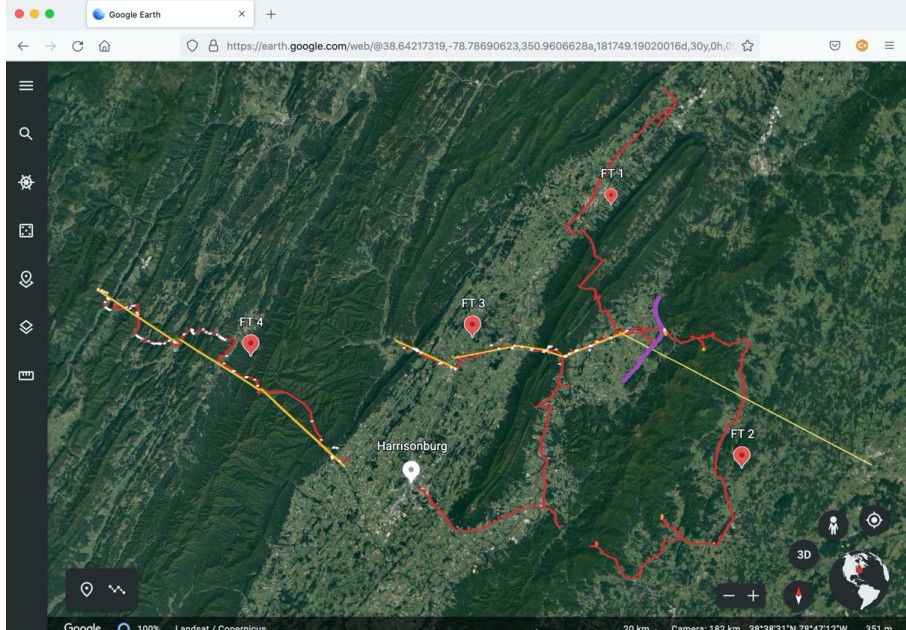

Figure 1. Screen image showing locations of web-based © Google Earth virtual
field trips in eastern West Virginia and western Virginia from the Mid Atlantic
Appalachian Orogen Traverse project; red lines indicate the paths of each field
trip (labeled FT1, FT2, FT3, FT4) and the yellow lines show the locations for
each cross section.
The SST field trips that encompass the Mid Atlantic Appalachian Orogen
Traverse (MAAOT) project typically consist of five all-day trips on weekends, and focus
on roadcuts or easily accessible outcrops along a generally east-to-west transect,
roughly perpendicular to the regional strike. Students work in teams to collect lithologic



and orientation data from each field trip site, and then spend time in discussions with
their colleagues and instructors to place the local outcrop data into a regional tectonic
context. In general, information from igneous and metamorphic rocks provides data for
the Grenville orogenic cycle, stratigraphic data provides the bulk of the evidence for
interpreting the Taconic and Acadian orogenies, and structural and orientation data
provides information for interpreting the Alleghanian orogeny. Some specific field
locations also provide data and information relevant to the breakup of the Rodinia or the
Pangaea supercontinents. The SST field trips are typically sequenced as follows:
Field Trip 1: This field trip functions as an introduction to Cambrian-Ordovician
sedimentary units of the Valley and Ridge geologic province in the context of the
rifting of Rodinia, formation of the Iapetan divergent continental margin, and the
subsequent Taconic orogeny. Students are introduced to methods of
stratigraphic data collection, analysis, and principles of basin evolution.
Field Trip 2: This field trip focuses on rocks of the Blue Ridge geologic province,
and students collect data on igneous and metamorphic composition and textures,
stratigraphic and sedimentological features, and structural/deformation features.
The tectonic context includes the Grenville orogeny, and two stages of the rifting
of Rodinia.
Field trip 3: This field trip progresses westward across the eastern part of the
Valley and Ridge geologic province along Rts. 211 and 259, effectively linking
with the northwestern end of Field Trip 2. Students primarily collect data on
stratigraphic features of Ordovician (Taconic orogeny and subsequent orogenic
calm) to Devonian (Acadian foreland basins) sedimentary rocks and later
structural/deformational features associated with the Alleghanian orogeny.
Field Trips 4 and 5: These field trips travel along Rt. 33 across the middle and
western parts of the Valley and Ridge geologic province, ending at the Alleghany
deformational front in West Virginia. The eastern end of the Rt. 33 traverse is
along strike with the western end of the Rt. 211/259 field trip. The Rt. 33 traverse
is divided into two field trips, as the distance covered, and the number of stops
visited, take up too much time for a single day's field trip. Students again collect
data on Paleozoic stratigraphic and structural features, and evaluate depositional
environments and tectonic events from the Cambrian through the Carboniferous
Periods.

On each of the first two field trips, student teams synthesize their field observations into
summaries of the geology and interpretations of the tectonic history of the region
traversed by each field trip. These tectonic synthesis reports are evaluated and
commented-on by professors, and returned to the students as iterative drafts of the final
tectonic summary report that student teams will produce at the end of the multi-week
project. Following the second and subsequent field trips, student teams draft interpretive





cross-sections along each field trip route, approximately perpendicular to the NNE-SSW
regional strike. Similar to the summary reports, these draft cross sections are each
evaluated and commented-on by professors, and returned to the students as iterative
drafts of the series of cross sections that collectively traverse the Appalachian orogen in
the Mid Atlantic region, which the students produce as part of their final project
deliverables (see Whitmeyer and Fichter, 2019 for more details on the project and
deliverables.) Through this iterative approach of collecting field data, drafting cross
section interpretations of the geology, and interpreting geologic data and models in a
summary report, students gain experience with data collection, interpretation, and
synthesis – key components of higher-order thinking in Bloom's taxonomy (Bloom et al.,
1956; Anderson et al., 2001.)
**2. The Transition to Virtual Field Trips**
Due to the COVID restrictions on travel, field trips for the Fall 2020 SST course had to
transition to a virtual format. There are several digital platforms that can be used to
display spatial and geologic data in an interactive format (Google Earth, ArcGIS, Unity
game engine, etc.); SST instructors used the web-based version of Google Earth to
host virtual field trips for the MAAOT, primarily for its ease of use and near universal
availability across a variety of computer hardware and mobile devices (see
https://www.google.com/earth/versions/ for more information.) Each of the standard on-
location SST field trips was redesigned as a Google Earth project that incorporated field
trip sites in the general sequence that would be visited during a standard on-location
field trip. The virtual Google Earth environment also facilitated the inclusion of extra field
locations for which there would not normally be enough time to visit during a typical on-
location weekend field trip.
The web-based Google Earth (GE) platform, though not as fully featured as the
downloadable desktop version of Google Earth Pro, has many features that make it
ideal for hosting interactive virtual geology field trips. Chief among these is that web-
based GE projects are hosted on the creator's Google Drive site, and thus can be easily
shared with students via a standard browser link (e.g. SST Blue Ridge Field Trip.)
Thus, in contrast to Google Earth Pro, web GE projects also can be interactively viewed
on mobile devices. Web GE projects can be designed to sequentially highlight stops
along a virtual field trip (Figure 2a) and can also include a full-screen title slide at the
start of a presentation (Figure 2b) to introduce the project and orient the user.



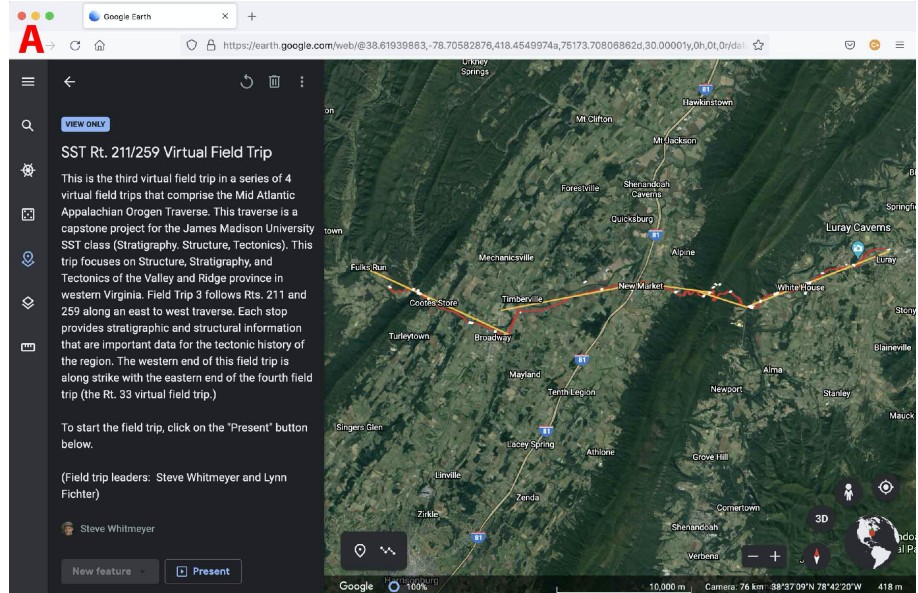

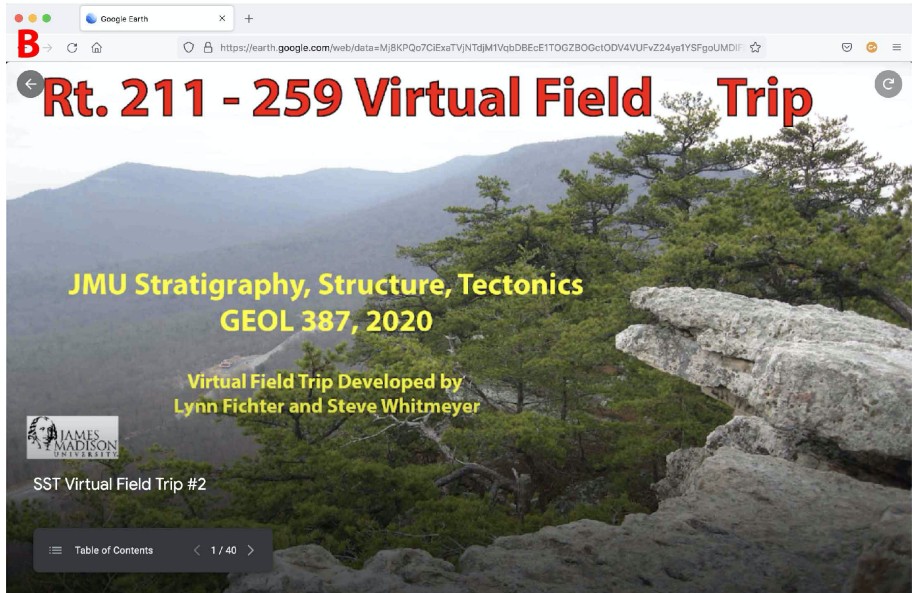

Figure 2. Screen images of web-based © Google Earth virtual field trip 3 from the
Mid Atlantic Appalachian Orogen Traverse project; A. Overview of the SST Rt.
211/259 Virtual Field Trip project in © Google Earth; B. Title slide for the Rt. 211 -
259 Virtual Field Trip in © Google Earth



Field trip locations can be highlighted with standard GE Placemark pins or with multi-
node lines, such that strike and dip symbols can be drawn at an outcrop location,
thereby replicating features of a standard geologic map (Figure 3a.) Each slide of a GE
project can be tailored to show a zoomed in bird's eye view of the location, or a
zoomable and rotatable Street View image of the actual outcrop (if Street View imagery
is available for that location; Figure 3b.) Each slide can incorporate a pop-up balloon
with descriptive text and an image carousel that can sequentially display up to eight
images or videos. Clicking on an image in the balloon will display an enlarged version of
the image, which is useful for showing annotations and details of outcrop features (e.g.
Figure 3c.) Short explanatory videos can also be included in the image carousel (e.g.
Figure 3d,) as long as the videos are hosted on YouTube and made available for public
viewing. Details on how the virtual field trips were designed and constructed in GE can
be found in Whitmeyer and Dordevic (2019), which highlights a virtual field trip across
the Blue Ridge Province in Virginia (Field Trip 2 of the MAAOT) as an example.

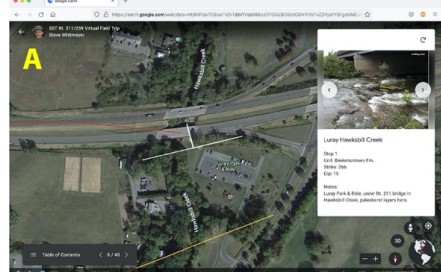
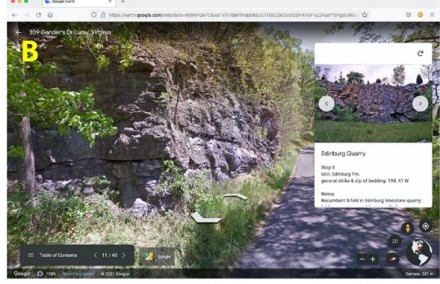
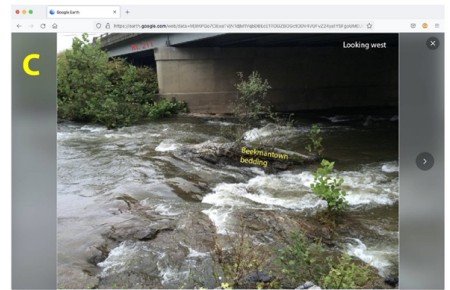
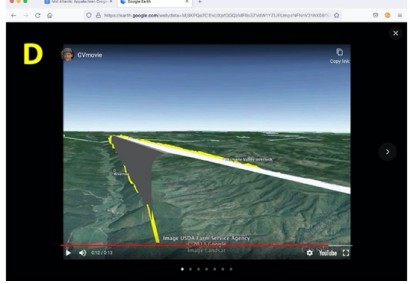

Figure 3. Screen images from web-based © Google Earth virtual field trips from
the Mid Atlantic Appalachian Orogen Traverse project; A. A virtual field trip site
that shows a birds eye view of the outcrop location with an oriented strike and dip
symbol drawn as a polyline in © Google Earth; B. A virtual field trip site that
shows a zoomable and rotatable Streetview image of the outcrop; C. An
annotated photo of a field site, shown as a enlarged image from the © Google
Earth slide carousel; D. A model of a regional anticline displayed as a popup
YouTube movie from the © Google Earth slide carousel.





The SST virtual field trips were conducted in a format that replicated the
organization of an on-location field trip, minus the driving from stop to stop. Students
and instructors (field trip leaders) assembled online using the Zoom virtual meeting
platform, and each participant had access to virtual field trip materials, including the GE
field trip project, PowerPoint files of supplementary materials, and other handouts as
PDF files. Instructors used the screen sharing mode of Zoom to virtually visit each GE
field trip site, show outcrop photos and other imagery in GE, and at some locations,
show more detailed "chalk talks" of images and background concepts using PowerPoint.
The concept of "chalk talks" derives from on-location field trips, where a field trip leader
would use a chalk board or a whiteboard to illustrate specific features or concepts
relevant to a given field location. For on-location field trips, SST students were provided
with a packet of paper handouts that consisted of annotated images and theoretical
models as supporting materials for the "chalk talk" discussions. Given the GE restriction
of only 8 slides in the image carousel, for the virtual field trips "chalk talk" materials were
provided as supplementary PowerPoint and/or PDF files that included images,
diagrams, and models.
On virtual field trips in SST, interactive explanations, discussions, and queries
about the geology of each site were conducted on Zoom in a similar format to on-
location field stops. Short breaks were taken every couple of hours between stops to
avoid Zoom fatigue, recognizing that down times in on-location field trips that occurred
during travel from stop to stop do not occur during virtual field trips. A longer lunch
break was also included, again replicating a traditional field experience (minus the visit
to the grocery store or restaurant.) Overall, even with frequent breaks, each virtual field
trip typically took less time than its on-location counterpart, likely due to the elimination
of the time needed for travel along the field trip route.
*2.1 Community Access to Virtual Field Experiences*
The transition of many undergraduate field experiences to virtual formats precipitated by
pandemic restrictions led to a grassroots effort by geoscience educators to assemble
examples of virtual field experiences in a publicly accessible web portal for use by the
community (Burmeister et al., 2020.) The National Association of Geoscience Teachers
(NAGT) Teach the Earth portal developed a new site, entitled "Teaching With Online
Field Experiences," to host an array of virtual field experiences and teaching modules,
ranging from introductory field trips to capstone projects, at virtual field sites around the
globe and beyond (https://serc.carleton.edu/NAGTWorkshops/online_field/index.html).
Four virtual field trips that encompass the MAAOT are included on the Teaching with
Online Field Experiences web portal as linked field experiences and educational
modules. Each of the virtual field trips is accessible via one the links below:
Field Trip 1: Stratigraphic Sequences of the Valley and Ridge Province
Field Trip 2: Virtual Field Trip to the Blue Ridge Province, Central Virginia



Field Trip 3: Rt. 211/259 transect
Field Trip 4: Rt. 33 transect
These field trip modules follow the general format of the NAGT Teaching with Online
Field Experiences portal, starting with a summary of the exercise (e.g. Figure 4, which
shows the webpage for Field Trip 3), followed by sections on the overall context of the
field experience, the educational goals, the technology requirements, useful teaching
notes and tips, and assessment strategies. Each module webpage includes a link to the
relevant GE field trip along with exercise handouts, supplementary materials ("chalk
talk" PowerPoint files), and other supporting documents.

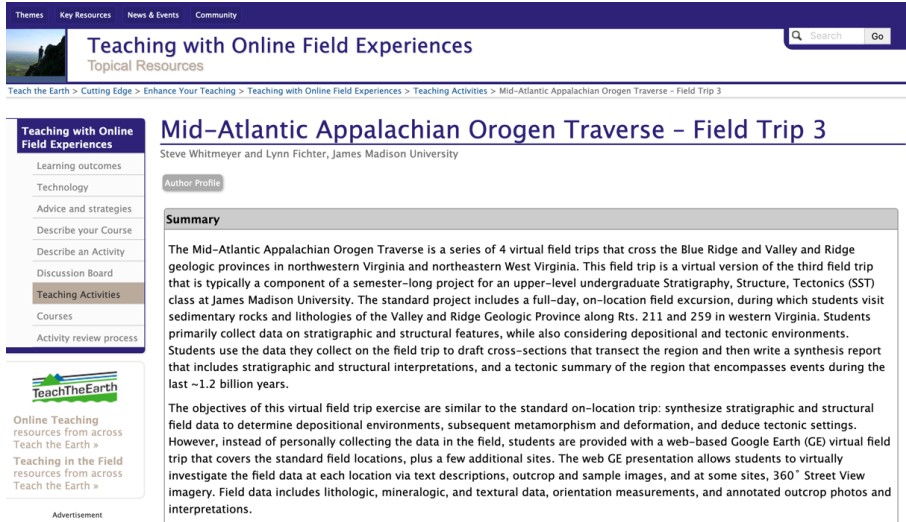

Figure 4. Screen image of the upper part of the NAGT Teaching with Virtual Field
Experiences webpage for the Mid Atlantic Appalachian Orogen Traverse - Field
Trip 3.
**3. Experiences With Virtual Field Trips**
*3.1 Instructor Experiences with Virtual Field Trips*
With the change to virtual interactions with students, SST instructors made significant
adjustments to their approaches to field-based teaching and learning. Several months of
development efforts were necessary to create the MAAOT virtual field trips in web GE
(as documented in Whitmeyer and Dordevic, 2021,) along with associated supplemental
materials. Fortunately, the instructors had collected field photos and videos from several
years of visiting the field trip locations with previous SST classes, and many of these
visual materials were included in the GE field trips. Similarly, supporting diagrams and
models had been developed in previous years and thus were available to include with
the virtual field modules as supplementary PowerPoint and PDF files.
Initial experiences with leading field trips virtually via the Zoom interface made it
clear that adjustments to teaching style and approach were necessary. On-location field





trips and educational field experiences typically highlight hands-on observations,
measurements, and field-based interpretations. Similarly, instructors in the field have
found it effective to ground their instructional approach in iterative cycles of encouraging
observation, followed by interpretation, followed by subsequent rounds of more detailed
observations and interpretations (e.g. Mogk and Goodwin, 2012.) Only after students
repeatedly have been encouraged to get as much information from each outcrop as
possible are they tasked with making bigger picture synthetic observations and
interpretations. Field tools and technologies have changed over the years, but the basic
approaches to field-based education have proven remarkably consistent (De Paor and
Whitmeyer, 2009.)
One of the challenges of virtual field trips is that what should be "observe and
discuss" can easily become "show and tell." Without the ability to read faces or body
language, observe students working the outcrop, or hold impromptu discussions, it is
easy to become disconnected from what the field experience is supposed to teach (e.g.
Petcovic et al., 2014.) Having at times lapsed into "show and tell" mode, the instructors
deliberately created protocols to avoid it, but it took time, effort, and attitude adjustment.
Instructors already had experience with online classroom lectures via Zoom, but often
that experience just encouraged slipping into a lecture format on a virtual field trip.
Experienced field instructors understand that field work has its own rhythms and
procedures, very different from the classroom (e.g. Mogk and Goodwin, 2012.) For
virtual field trips the challenge is to create an interactive learning experience for the
students in a less familiar format. The process of redesigning field trips for a virtual
environment started with instructors re-visiting an outcrop and systematically and
deliberately analyzing everything that typically occurs, from getting out of the vans to
getting back in. With that mind-set recreated, significant time (hours to days) was
devoted to recreating each field site virtually, as there were many practical problems to
solve, including assembling detailed field photos and diagrams, some of which were not
available and had to be collected.
*3.2 Structural Analyses on Virtual Field Trips*
Structural analyses on SST field trips initially focus on characterizing lithologies and
recognizing where in the stratigraphic sequence an outcrop is positioned, in addition to
knowing where the outcrop is located geographically. Secondly, students need to record
the orientations of planar fabrics, such as bedding or foliation, and recognize broad fold
patterns and geometries from changing dip amounts and alternating dip directions.
Thirdly, lineations and other outcrop-scale deformation fabrics (e.g. slickenlines,
asymmetric porphyroclasts, etc.) are important to recognize and measure, where
apparent.
The virtual field environment presents several challenges for collecting
structurally-related outcrop information and data. Identification of rock types and



differentiation of lithologic units can be difficult with static images. Replicating orientation
measurements online is a significant challenge, although virtual compasses do exist as
components of some virtual outcrop experiences (e.g. Masters et al., 2020.) Our
approaches to virtual field trips centered on providing outcrop imagery at multiple scales
and in different formats (e.g. static outcrop photos, dynamic Street View images; Figure
5a,) often with annotations to highlight important features (Figure 5b.) Instructors used
this imagery during Zoom discussions to iteratively encourage students to make ever
more detailed observations of an outcrop, making sure that students obtained the
salient lithologic and structural information that would aid in their subsequent tectonic
interpretations.
Outcrop orientation measurements can be extremely difficult to facilitate in a
virtual environment, and the experience of using a virtual geologic compass is currently
ineffectual with a web-based platform like Google Earth. Thus, the approach in the
MAAOT field trips is to provide orientation data in the pop-up balloons associated with
stops that featured bedding, foliation, and/or lineation information (e.g. the text in the
pop-up balloons of Figures 3a, 3b, 5a.) This is clearly not the same pedagogical
experience for students as using a physical geologic compass (e.g. Brunton Pocket
Transit) to take their own measurements on an outcrop, but the instructors accepted
that this was not a skill that could be effectively replicated virtually.
Key deformation fabrics that are visible on an outcrop can be highlighted virtually
via images, and an advantage of the virtual environment is that photos can include
annotations that explain the relevant structural interpretations of a particular feature. For
example, ductily-deformed porphyroclasts that display asymmetry can be used to
determine the direction of movement that occurred during a faulting event (Passchier
and Simpson, 1986.) Annotations on an outcrop photo can clearly demonstrate to
students the appropriate way to interpret these features, as with the complex sigma
porphyroclast in Figure 5c that displays a top-to-the-left sense of movement. In addition,
virtual images and animations can illustrate or model structural features that are at a
regional scale - much larger than can be viewed at a single outcrop (e.g. the kilometer-
scale anticline modeled in Figure 2d.) Instructors often attempt to model these larger
structures for students while on-location at a key outcrop using verbal descriptions or
hand waving, but they lack the ability to figuratively "step back" and illustrate the bigger
picture. The ability to take a regional view of large features, and if desired display a
model of them, is a distinct advantage of the virtual environment.

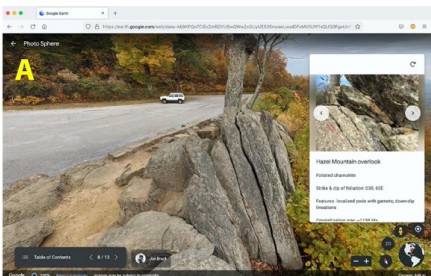
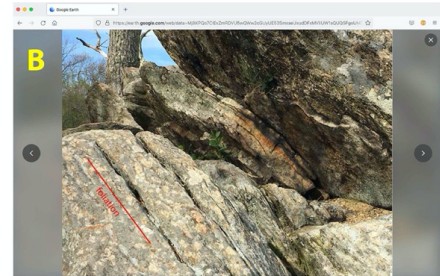

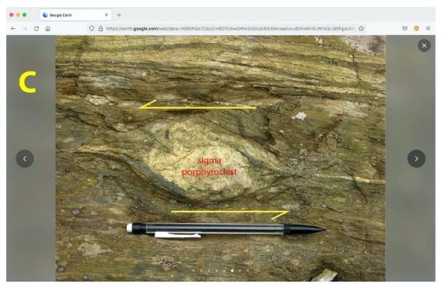
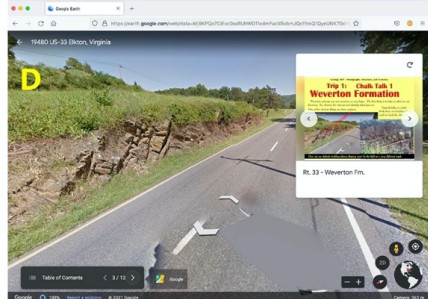

Figure 5. Screen images from web-based © Google Earth virtual field trips from
the Mid Atlantic Appalachian Orogen Traverse project; A. A Street View image of
the Hazel Mtn. Overlook site from FT2, positioned to look along strike of foliation;
B. An annotated photo of the same outcrop as A., highlighting the foliation; C. An
annotated photo of a complex sigma porphyroclast from the Garth Run site of
FT2; D. A Street View image of the first field trip site of FT1 on Rt. 33 in western
Virginia.

*3.3 Stratigraphic Analysis and Basin Evolution on Virtual Field Trips*
Field-based stratigraphy and basin analysis require a different approach from analyzing
structural features. Unlike tectonic structures (folds, faults, slickenlines, etc.,) which are
often apparent on an outcrop, tectonic basins are not visible at the outcrop scale; they
are too large. In addition, depositional environments are interpretations built on a
hierarchy of observations, none of which are intuitively obvious. The goals of field-based
stratigraphy/basin analysis are to use bottom-up empirical data to construct a tectonic
basin interpretation, or use theoretical first principles and models to make
interpretations of outcrop observations, and move freely back and forth between both
approaches. The approaches to field-based stratigraphy and basin analysis in SST
previously have been presented in detail (Fichter et al., 2010; Whitmeyer & Fichter,
2019.) The paragraphs that follow highlight how these approaches have been adjusted
and modified for the virtual environment.
Theoretical principles and models of stratigraphy/sedimentation and basin
analysis are developed in SST classroom lectures and discussions, but commonly
these topics have not been fully explored prior to the earlier field trips in the MAAOT. In





addition, the practical field skills of recognizing and identifying sedimentary structures
(e.g. is it trough, planar, or hummocky cross stratification?) and stratigraphic sequences
(Bouma, hummocky, point bar, etc.), and drawing strip logs must be learned through
practice. Even if these concepts have been presented in the classroom (usually from
drawings and pictures) students typically have to relearn them on the outcrop, via one-
on-one, back-and-forth conversations that take place while looking at the rocks. The
challenge of developing the SST virtual field trips was to reproduce these experiences
in Zoom, using GE-based presentations and PowerPoint "chalk talk" mediums, where
conversations are often fragmented or non-existent. Unlike many classroom lectures,
field trips are interactive environments, and when it is difficult to discern facial
expressions or body language, creating an interactive learning environment requires
different strategies and approaches.
As an example, the first stop of Field Trip 1 in the MAAOT is a small roadside
outcrop of weathered Weverton Formation (Figure 5d) that embodies many of the
challenges of investigating virtual field sites. Examination of an outcrop on an SST field
trip starts with geography: "Where are we?" Constructing basin interpretations requires
data from many outcrops across a wide region, and it is important for students to know
the spatial relationships between the outcrops. This is a practical problem even on an
on-location field trip; many students just blindly travel from stop to stop without keeping
track of their geographic locations. The GE component of the virtual field experience
makes it easy to show the location of an outcrop within the region, which helps students
conceptualize the regional geologic context.
Analysis of the Weverton Fm. outcrop proceeds using the GE Street View image,
by virtually walking past the outcrop, zooming out, zooming closer, and viewing it from
different angles. In an on-location field trip this first phase of observation involves many
prompts: "Go look at the outcrop!" "Ok, what did you see?" "Did you look for this and
this; did you see this?" "Go look again." "Here, let me show you something; what do you
make of that?" This incorporates as many back and forth iterations as are necessary,
integrating across many scales of observation, while at the same time building a
stratigraphic, basin analysis, and tectonic story. At a virtual field site, with or without
Street View, this also requires an encyclopedia of detailed and annotated photos.
An important element of these initial observations is separating out structural
features, metamorphic overprinting, weathering phenomena (e.g. liesagang), etc. Each
of these is addressed individually as an outcrop datum, but the initial parsing is an
important component of SST; again, this is aided by using supplemental photos that
emphasize different features. Outcrops are not always examined and discussed with the
same hierarchy or order of investigations; sometimes structural analyses come first,
sometimes stratigraphic features are emphasized. When stratigraphic features are the
focus, many scales of observation and different views are necessary. The outcrop is
initially viewed from a distance, with prompts such as: "What do you see?" "Are these





carbonates or clastics?" "Where is bedding and how is it oriented?" "Can you say
anything about texture?" "What is the QFL? (e.g. relative content of quartz, feldspar,
and lithic fragments)" Many of these questions are presented as hypotheses and involve
back and forth conversations, refining the students' outcrop observations.
More detailed views are next, with focused photographs of representative parts
of the outcrop that include annotations, which highlight bedding, sedimentary structures,
textures, etc. Students are asked probing questions in a dialogue that develops the
necessary theoretical background, while sharpening their observation skills. However, it
is challenging for students to learn to recognize features like hummocky stratification
from a photograph. Thus, the quality of the photos is important; they have to be clear
and unambiguous, which often necessitates multiple views of a feature. To facilitate
this, the instructors revisited many MAAOT outcrops prior to the start of the Fall 2020
semester, in order to get high resolution pictures in the best lighting conditions.
Another practical problem is the challenge of getting students to talk and interact.
This can be challenging in an in-person classroom setting as well, but the virtual Zoom
medium unfortunately facilitates reticence from students. Strategies to mitigate this are
not that different from being on an outcrop, and include asking a question and letting the
silence hang there until someone addresses it, or reframing the question, or doing a
mini-quiz. Taking the time to get conversations started is necessary, and the key is to
keep the conversations going throughout the field trip. As the field day progresses
students get more comfortable with the discourse, as long as an interactive discussion
framework is initiated early in the trip.
The culminating empirical activity is for students to draw a strip log from an
outcrop photo, or a sequence of photos as necessary. A successful strategy starts with
thoroughly discussing the stratigraphic section under consideration (specific images
were obtained for this purpose,) making preliminary observations, and initiating a
dialogue about what is observed. This interactive discussion is slow and deliberate.
Then students draw their own strip logs from a combination of what they have observed
and information they have developed via the discussions. At this point on an on-location
field trip everyone would lay their strip logs down on the ground for group examination,
featuring prompts from the instructors, such as: "What do you like; what don't you like;
what would you do differently?" "What is missing?" "What would make it better?" This is
awkward to accomplish virtually, although one approach is for students to hold their
drawings up to their laptop or mobile device cameras for viewing by the group. This can
work in a small class with a few students, but is more time consuming with two or three
dozen students. Eventually, an instructor's strip log was displayed as an example,
followed by comparisons with the students' work and questions, etc. Students then were
tasked with redrafting their strip logs. This progresses through as many iterations as are
necessary, with the primary goals of building observational and interpretive skills.





The final step is to move to multiple layers of interpretation, which become
progressively more abstract and more theoretical. This is where a virtual "chalk talk" is
valuable. In an on-location field trip theoretical interpretations are presented with
posters ("chalk" boards) tacked to the sides of vans. This can be problematic in lousy
weather, or in a large class where students on the distant edges of the group have
trouble seeing and hearing the discussion. Virtual chalk talks on Zoom using
PowerPoint slides obviates this - everyone has the same access and opportunity to
interact. At the higher interpretive levels discussions become more and more
theoretical, applying models initially presented in classroom lectures to the outcrop data.
Initially, the theoretical models probably don't have much relevance to the students, but
because the chalk talks can easily transition to lectures with high quality illustrations as
necessary, learning can be effective. As the stops accumulate throughout the field day,
and these theoretical models keep reappearing and building on each other, they
become familiar and increasingly more relevant to the students.
**4. Survey of Student Experiences with In-Person vs. Virtual Educational Formats**
Historically, the geosciences have been largely field-focused (e.g. Himus and Sweeting,
1955), and undergraduate curricula have traditionally incorporated a significant
component of field-based learning (Whitmeyer et al., 2009; Mogk and Goodwin, 2012.)
This field emphasis has been used for many years to recruit students to the discipline
that have an affinity for, and appreciation of, the outdoor environment. An ongoing
challenge in geoscience disciplines is to increase access and inclusion for all students
(Bernard and Cooperdock, 2018; Ali et al., 2021; among many others,) yet field-based
learning experiences can present a significant barrier to those efforts (e.g. Clancy et al.,
2014; Giles et al., 2020.) Disability access to field environments is a growing concern
among geoscientists and geoscience departments (Carabajal et al., 2017; Whitmeyer et
al., 2020,) especially with regards to recruitment and retention of students in
geoscience-related fields (Baber et al., 2010; LaDue and Pacheco, 2013; Stokes et al,
2015; Pickrell, 2020.) Virtual field experiences are one potential solution to inaccessible
field experiences, but little data exists on academic growth during virtual field
experiences and how that growth compares to in-person field learning.
With these things in mind, an online survey was developed. The survey was sent
to SST students that had participated in the virtual field trips for the MAAOT in Fall
2020, as well as to SST students from previous years that had participated in traditional
on-location field trips. The survey included questions that addressed student
preferences for in-person or virtual field experiences, self-evaluations of academic
growth across a range of topics relevant to the SST course, and questions that
addressed student disabilities in the context of field access and inclusivity. Details of
survey questions are available in Appendix A.





Responses to the survey were received from 11 students that participated in
virtual field experiences in the Fall 2020 semester, and 21 students that participated in
on-location field trips from the SST course in previous years. Data were collected
anonymously via an online survey instrument using Survey123 through ArcGIS Online,
with IRB approval obtained from JMU. Survey data was aggregated across all
responses, or aggregated within two groups: students that participated in virtual field
experiences, and students that participated in on-location field experiences. All data
was anonymized to remove any information that could facilitate identification of
individual respondents. The results were then organized into three themes: preferences
for in-person vs. virtual field experiences, disability and field access, and a comparison
of academic growth between in-person and virtual field learning.
*4.1 Student Preferences for Virtual vs. In Person Learning Experiences*
Prior to Fall 2020, the lectures, labs, and field trips in the SST course were all
conducted in-person and on-location in the field. None of the students that took SST
prior to Fall 2020 had experience with virtual classes or virtual field trips, outside of the
occasional use of a virtual platform like Google Earth to illustrate regional to global scale
topographic or geologic phenomena. Not surprisingly, students that took the SST
course prior to 2020 did not indicate a preference for virtual learning, although a few
students recognized the potential value of hybrid (some combination of virtual plus on-
location) experiences (Figure 6a.)

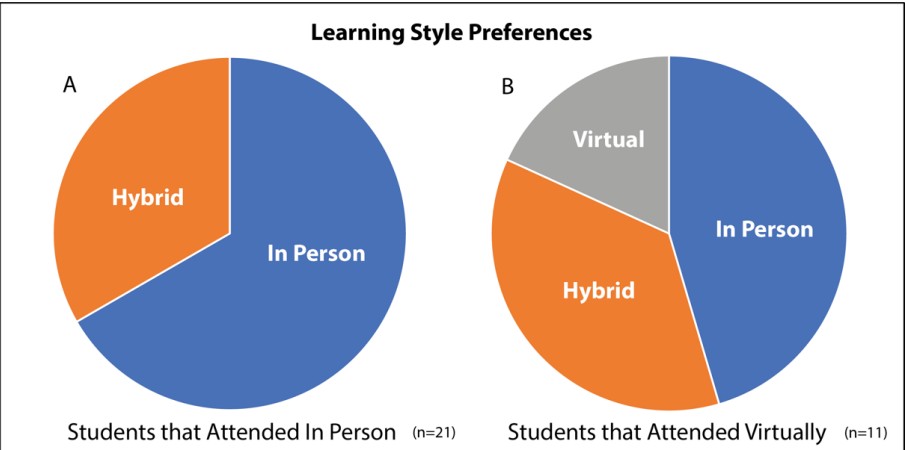

Figure 6. Charts of learning style preferences from student survey; A. Learning
style preferences from students that attended SST classes and field trips in
person, with no preferences for virtual learning style indicated; B. Learning style
preferences from students that attended SST classes and field trips virtually, with
a greater preference for hybrid and virtual learning styles.




Some students that experienced virtual learning and virtual field experiences in the Fall
2020 SST course likewise indicated a preference for in person experiences; however, a
majority of these students indicated a preference for hybrid or virtual learning
experiences (Figure 6b.) In addition, most of the Fall 2020 students that attended SST
virtually indicated that they had some concerns about virtual field trips prior to
experiencing them (Figure 7.) However, Figure 6b suggests that many of these students
gained an appreciation for virtual field experiences by the end of the course.

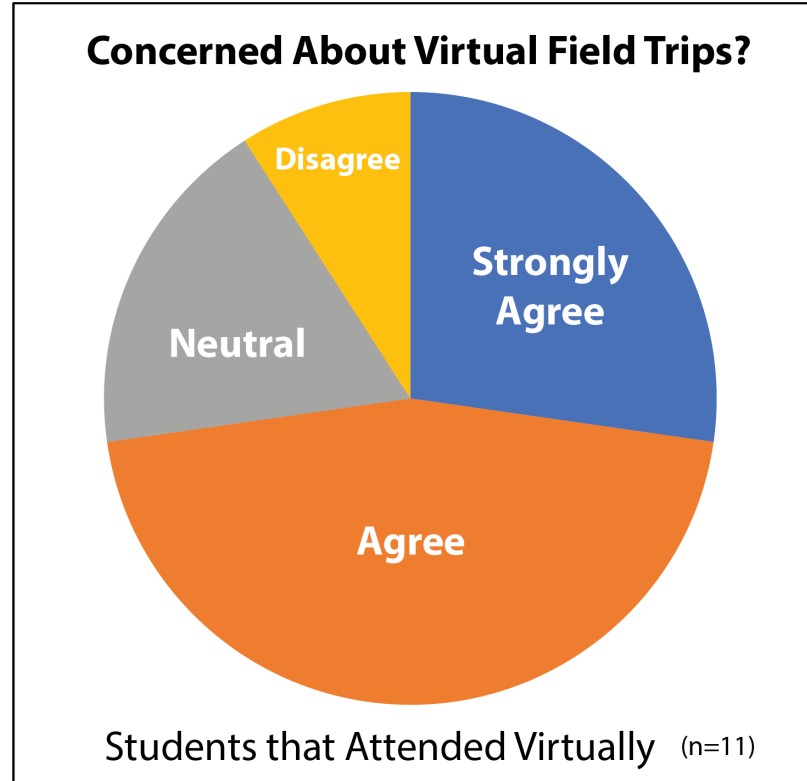

Figure 7. Chart of responses from students that attended SST virtually on
whether they were concerned about participating in field trips virtually.
For many students virtual field experiences were not as satisfying as being
physically at an outcrop, as noted in the following response from a student that attended
SST virtually:
*"While I feel as though I have missed out on an important [field] experience by*
*taking SST online…"*
However, that response continues with:
*"…I feel I learned more than I would have because of my ability to re-watch*
*lectures and go back to the [virtual] field trips."*





This response is representative of several student responses that noted the advantage
of reviewing and revisiting virtual field trips and field sites after an initial experience. This
includes several students that attended on-location field trips, who indicated a curiosity
about, and an awareness of, the potential for virtual field experiences. Some examples
of these responses include:

*"I took all in-person geology courses prior to graduating, so I was never given the*
*option to take any field trips virtually, but I wish I could have seen how they may*
*have worked, and what software was used."*


*"The virtual field trips in google earth are very well done and I think those things*
*are helpful."*


*"...I have never attended an online field trip, so I am unfamiliar with them. It would*
*be nice to have the opportunity to catch anything I might have missed during field*
*trips [due to] loud cars, not [standing] close enough to the speaker, or having to*
*sit out on a few steep outcrops."*

The response above also highlights the inclusivity of virtual field experiences, where
every student has an equal opportunity to examine and investigate each outcrop and
participate with other students and instructors, regardless of physical ability or proximity
to ongoing discussions. Accessibility aspects of virtual field experiences are discussed
in more detail in the section that follows.

*4.2 Student Views on Disabilities and Field Access*
Survey results indicate that a majority of SST students agreed that students with
disabilities may be deterred from majoring in the geosciences due to the expectation
that fieldwork is a necessary component of upper-level courses (Figure 8.) Many SST
students, across both learning modalities (in-person and virtual,) indicated an
awareness of challenges and issues associated with disability access in field settings.
As one student noted,

*"...the geosciences in general have a stereotype of being the science of the*
*rugged outdoorsman, and that deters people with disabilities."*


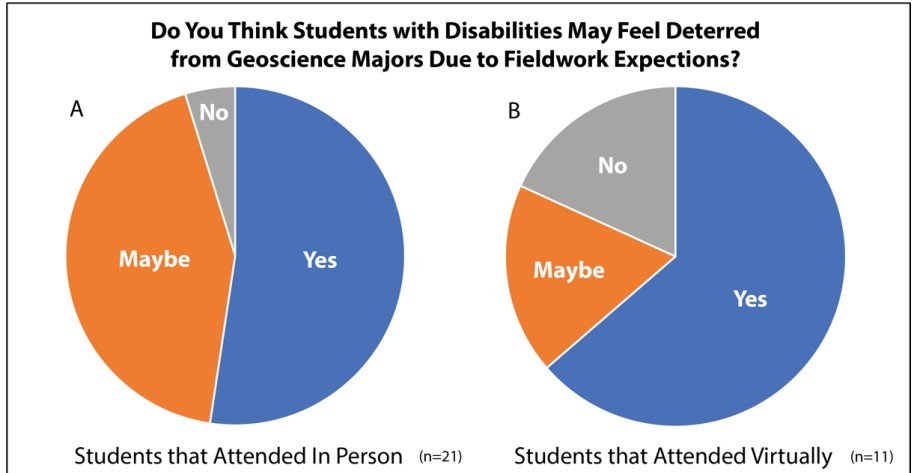

Figure 8. Responses from students of both in person and virtual modalities on
whether they thought students are discouraged from majoring in the geosciences
due to a fieldwork requirement in undergraduate curricula.
Table 1 contains narrative responses from the student survey that reflect disability
access and inclusion issues for field trips, including those in the SST course. Several
SST students dealt with accessibility challenges during the on-location field trips and
indicated that they would have welcomed the option of viewing and investigating
outcrops virtually. Students that participated in virtual field trips also indicated an
awareness of field access issues for students with disabilities, as highlighted in the last
few responses in Table 1. Regardless of whether students had experience with virtual
field trips, there was recognition that issues like navigating topographic relief to see
outcrops close-up, or just getting in and out of vans multiple times during a trip,
presented challenges for some students. Virtual field experiences were seen as a viable
alternative by many students, regardless of whether they had experience with virtual
modalities.



| Student Comments on Disability Access and Inclusion in the Field |
| --- |
| |
| Comments from students that took SST in person |
| "Physical challenges such as knee/joint/etc. pain as well as heart issues, affected my ability to fully interact with the outcrops (especially ones that required foot travel)." |
| "I had a knee injury that prevented me from standing for long periods of time, climbing up or down to see certain outcrops, and needing help when taking measurements like strike and dip because my balance was not exactly up to par. I did not get to see every outcrop or help take measurements, and I felt that I was more of a burden to my group then a help overall because of this." |
| "Many field trips … involved climbing very steep inclines which worried me with some of my health issues. If you didn't climb, you missed out." |
| |
| Comments from students that took SST virtually |
| "…the virtual field trips offer an opportunity for students with physical limitations to participate … it is a good option for them, but the other students need the in person experience out in the field as well." |
| "…if it were not for covid, I would not have been able to really participate in field trips." |
| "I can definitely see how disabilities could make physical field work difficult, but the online presentation of the material is very useful and efficient…" |
| "The google earth features with field trip info at each stop … is certainly … accessible and helpful to those with disabilities in most cases." |

Table 1. Responses from the student survey that discuss disability access and
inclusion issues for field trips. Responses are grouped according to modality of
learning environment (in-person or virtual.)
Student responses also highlighted the potential for technological solutions to
augment field experiences. Some students were made aware of the potential for mobile
communications devices to augment field experiences for disabled students via a
student presentation that highlighted ongoing research (Atchison et al., 2019;
Whitmeyer et al., 2020.) The responses below were from students that attended SST in-
person, but recognized the potential of technology for improving field access:
*"I saw the use of ipads and video chats to help those with physical disabilities*
*that may not be able to visit certain onsite locations."*





*""…the student had tested a novel system for broadcasting outcrops which were*
*inaccessible to students with disabilities through livestreaming on an ipad or*
*similar technology. Seemed like it had a lot of potential!"*
These responses highlight the possibilities for enhancing accessibility in the field and
suggest ways for improving inclusivity for SST and other geoscience courses, as a
hybrid approach to virtual and in-person learning.
*4.3 Student Perception of Academic Growth during the SST Course*
Students were asked to self evaluate their academic growth from the beginning to the
end of the course. Students used a scale of 1 (little academic growth) to 10 (most
academic growth possible) to evaluate their overall academic growth during the
semester, as well as growth in key topics in the general areas of stratigraphy, structure,
and tectonics (Table 2.)

| Academic Growth in Key Topics of Stratigraphy, Structure, Tectonics (SST) Course | | | | | |
|---|---|---|---|---|---|
| | Students that Took the Course In-Person | | Students that Took the Course Virtually | | Discrepancy in |
| Topic | Mean of Responses | Range of Responses | Mean of Responses | Range of Responses | Means of Responses |
| Identifying and understanding depositional environments | 6.90 | 3 - 10 | 6.18 | 2 - 10 | 0.72 |
| Constructing strip logs | 6.90 | 3 - 10 | 5.27 | 1 - 9 | 1.63 |
| Ability to apply the geologic time scale on field trips | 7.43 | 4 - 10 | 6.82 | 3 - 9 | 0.61 |
| Interpreting cross sections and identification of geologic structures | 8.05 | 5 - 10 | 6.73 | 3 - 9 | 1.32 |
| Evaluating structural concepts and deformation | 7.14 | 2 - 10 | 7.09 | 4 - 10 | 0.05 |
| Tectonic Interpretations of Rocks and Minerals | 6.43 | 3 - 9 | 6.09 | 4 - 9 | 0.34 |
| Interpreting and applying the Wilson Cycle | 6.95 | 3 - 10 | 5.73 | 2 - 10 | 1.22 |
| Understanding tectonic events through time | 7.29 | 3 - 10 | 7.00 | 4 - 9 | 0.29 |
| Overall academic growth | 7.57 | 5 - 10 | 6.64 | 3 - 8 | 0.93 |

Table 2. Student survey responses highlighting self-evaluation of academic
growth from the beginning to the end of the Stratigraphy, Structure, Tectonics
(SST) course. Responses are grouped by whether the students took the course
in-person (*n=21*) or virtually (*n=11*.) Key topics highlighted include those with a
stratigraphic focus (in yellow), those with a structural focus (in blue), and those
with a tectonics focus (in green). Academic growth is reported on a scale of 1 -
10, where 1 = little academic growth and 10 = the most academic growth
possible; means of responses and ranges of responses are indicated.
In all categories students that took the course in person reported higher mean scores
than students that took the course virtually. In general, stratigraphy topics displayed a
greater discrepancy in mean responses between students that attended in person and
students that attended virtually. However, the topical categories that show the greatest
discrepancies between in person and virtual attendance encompass all three general
areas: strip logs (deviation of 1.63; stratigraphy), cross sections (deviation of 1.32;
structure), and the Wilson Cycle (deviation of 1.22; tectonics.) It is worth considering





that these three categories represent topics that require synthesis of data in the
preparation of summary diagrams, interpretations, or models. This disparity between
modes of attendance in students' perceptions of their abilities to synthesize data may
also be reflected in the relatively significant discrepancy (0.93) in their evaluations of
their overall academic growth during the semester.
Student perceptions of their academic growth during the SST course reflected
classroom, laboratory, and field learning environments. Thus, the deviations between
the higher self-reporting scores for students with in-person attendance and the lower
scores for virtual attendance do not only reflect on-location vs. virtual field experiences.
However, several topics that directly address field-oriented learning (constructing strip
logs, ability to apply the geologic time scale on field trips, interpreting cross sections and
identification of geologic structures, understanding tectonic events through time)
indicate that students that participated in virtual field experiences were generally less
confident of their academic growth in field-focused learning than students that
participated in on-location field trips. Several factors likely contributed to this result.
First, the SST instructors have many years of experience with on-location field
trips and have fine-tuned the MAAOT trips over the course of several years to maximize
the student experience. In contrast, Fall 2020 was the first semester in which the field
experiences were fully virtual, and it is likely that the student learning environment was
less effective and less positive as a result. Many SST students seem to look forward to
the field trips as highlights of the course, and in 2020 many students expressed
disappointment or even apprehension (e.g. Figure 7) that the field trips would have to
switch to virtual delivery and participation. These apprehensions are highlighted in some
qualitative responses to the student survey; for example:
*"As someone who would not consider themselves to have a severe disability,*
*[the SST course] still took a huge toll on me both physically and mentally."*
*"We are told that a geologist is only as good a geologist as the amount of*
*geology they see and a lot of people with disabilities can't see all of the things*
*able-bodied people can."*
Reduced enthusiasm for the virtual field component of the course may have resulted in
less effort by the students. However, apprehension for on-location field trips on the part
of students with mobility challenges or other environmental concerns may have been
alleviated once students gained experience with virtual field trips. In addition, it is likely
that the general frustrations of both faculty and students with the restrictions imposed by
the COVID pandemic had negative effects on the academic learning environment as
well as on general living conditions. These effects are hard to quantify but were certainly
experienced by the authors and expressed to them by many students during the Fall
2020 and subsequent semesters that were impacted by the pandemic.





**5. Discussion**
Many of the challenges faced by instructors with the switch to virtual field experiences
revolved around determining the most effective ways to accomplish traditional field
learning goals (e.g. Mogk and Goodwin, 2012; Petcovic et al., 2014) within a less
familiar virtual environment. Engaging students in a dialogue can be challenging in a
virtual environment where students may or may not have web-linked video cameras
turned on, and may have other distractions going on concurrently in their home
environments. Asking students to focus on virtual images of outcrops to discern salient
features is not the same as tactile investigations of an outcrop in the field. Important
outcrop details usually need to be highlighted in an image through annotations (e.g.
Figure 2c) or explained in a video. This is not the same experience as directing students
to examine an outcrop to find these features for themselves. However, if an effective
dialogue can be established between students and instructors in the virtual
environment, many of the same interpretation and synthesis goals can be achieved
through probing questions and repeated directed observations. One advantage of virtual
field trips is that supporting diagrams, models, and other materials are immediately at
hand and can be easily displayed (e.g. Figure 2d) and annotated in real time by
instructors and students. Similarly, process-based models that sequentially change
through time can be easily displayed virtually, which would be more challenging to show
and discuss on location in the field. These and other relative advantages and
disadvantages of virtual field experiences vs. on-location field trips are discussed in
more detail below.
*5.1 Pedagogical Advantages and Disadvantages of Virtual vs. On-Location Field*
*Experiences*
On-location field experiences have been the traditional format for field-based education
for many years, and virtual field experiences are typically evaluated in comparison to
on-location trips. If the statement attributed to Herbert Harold Read that "The best
geologist is the one that has seen the most rocks." (Young, 2003, p. 50) has merit, then
virtual field experiences would seem to have inherent weaknesses that could be
challenging to overcome, some of which are readily apparent, such as:
1.  The tactile components of on-the-outcrop investigations. On virtual field trips
students do not experience their own self-directed examinations of the rocks
(minerals, fabrics, structures,) which can inhibit observationally-grounded
geologic interpretations. In addition, field skills, such as using a hand lens for
detailed observations or taking outcrop measurements with a geologic compass,
are not effective in a virtual environment, and thus students don't have the
opportunity to practice and refine these field-oriented skills.



2. A clear appreciation of the spatial dimensions of the region and the relative
locations of outcrops. Virtual experiences via Google Earth are effective in
showing birds-eye or regional views of a field trip area, but the actual separation
and distance between each outcrop is more easily grasped when physically
traveling from location to location on the ground, whether walking or driving.
3. Learning safety in the field. During on-location field trips instructors spend
significant time and effort highlighting outcrop safety. MAAOT field trips
incorporate many outcrops that are roadcuts along busy highways, and many of
these outcrops are steep or subvertical and tower above the students.
Throughout an on-location field trip, participants are encouraged to wear
reflective vests, and instructors are constantly yelling "Rock!" or "Car!" to
encourage safety on the outcrop; this sense of awareness of one's surroundings
and physical environment cannot be experienced virtually.
4. A sense of appreciation and enthusiasm for the natural world. Historically, one of
the drivers for recruitment in the geological sciences is the sense of wonder and
excitement that students obtain from being physically present in awe-inspiring
natural settings (e.g. Petcovic et al., 2014.) This emotional connection with the
real world is not present in virtual electronic environments.
However, virtual field trips offer some distinct advantages, as highlighted below with
reference to the MAAOT field trips.
1. On virtual field trips it is not necessary to visit outcrops in the order dictated by
geography and the local road network. In the region of the MAAOT it is possible
to visit many formations in stratigraphic order, but that is not always the case in
other regions. In areas where outcrops are not chronologically sequenced, field
locations can be mixed and matched, using Google Earth to keep students
geographically oriented.
2. On an on-location field trip each outcrop has to be examined for every piece of
stratigraphic, structural, and tectonic evidence while at the outcrop. This tends to
make field notes complex and chronologically disjointed, and can break up the
rhythm of interpretations. On a virtual field trip a series of outcrops can be visited
to understand the structural details, then revisited to focus on stratigraphic
details, and then revisited again for basin analysis and tectonics. It can take more
time, but this approach can facilitate better organization of the information by
students.
3. An on-location field trip cannot easily incorporate observations from related but
distant outcrops of the same formation that illustrate variability or regional facies
changes. On a virtual trip, stops at different locations that feature the same rock
unit can be visited sequentially as a group to cohesively present the data
available, and investigate changes across distances.





4.  Because the MAAOT virtual field trips incorporate PowerPoint supplemental files it is possible to include many images that might not be easy to examine on location at an outcrop. For example, environmental interpretations of the Juniata and Tuscarora Formations (Field trips 3 and 4) can be facilitated and enhanced by using pictures of contemporary tidal flats and beach/barrier island systems. Or, for the Acadian Catskill clastic wedge, atmospheric circulation models and paleo positions, as well as paleontological evidence, can be helpful for reconstructing possible environmental conditions during deposition.

5.  In virtual field trips, all of the students get the same amount of time and opportunities to examine an outcrop. In contrast, with large classes and small outcrops, in on-location field trips instructors cannot be sure that everyone has had ample time on the outcrop to see all of the salient details. Similarly, students may not have had equal opportunities to discuss the outcrop with the instructors. In addition, some outcrops are physically challenging to get to (e.g. the necessity of climbing steep or unstable slopes to see an outcrop.) With virtual field trips all students have equal access to an outcrop.

6.  Students can easily revisit virtual field trips and field locations for quick reminders and reviews, as long as the virtual field trip files are made available during and after the instructor-led field trips. This can be an effective mechanism for student teams to revisit MAAOT field trip sites while they are working on their cross section interpretations and synthesis reports.

7.  The GE virtual format provides the opportunity to take field trips to distant locations that might not otherwise be feasible or practical for on-location field trips. As the library of high quality virtual field trips accumulates (e.g. NAGBT's Teaching With Online Field Experiences site) it will be possible to take students on field trips to many places in the world that otherwise might not be accessible.

*5.2 Student Perceptions of Field Experiences*

Survey results indicated that students that took SST in-person generally were unaware of virtual field experiences. For students steeped in the tradition of field-based geology, it is not surprising that they did not envision options for virtual or remote field experiences. However, several student responses from the survey indicated the perceived importance of on-location field trips, while also recognizing the potential for a hybrid approach that incorporated both on-location and virtual features. Survey responses from students that noted specific benefits to a combined hybrid approach are highlighted below.

1. Field accessibility

*"Offering more virtual options to students in the future, even if most of the class chooses to do in-person versions. I think most students, like myself, prefer in-person field trips, but I can see how it may be hard for some students to do that."*






*"For outcrops that I was (and other individuals were) unable to traverse to/focus*
*on, incorporating a 'virtual' aspect, similar to what's being offered now, would've*
*been useful to allow us to see the outcrop without having to forgo the*
*experience/knowledge."*

2. Revisiting field sites:
*"…a virtual option for outcrops, … where I would be able to catch up on the*
*material I was unable to [see], would be vastly useful."*

*"Having a resource of a digital version of the [field] trip, with some key photos*
*and points of the stop to assist in aligning personal notes with the stops would*
*have been a helpful re-enforcer."*

3. Incorporating modern mobile technologies to enhance inclusivity
*"Virtual field trips in addition to physical/in-person ones – i.e., having someone*
*with a cellular-enabled iPad come along on the field trips to stream video back to*
*anyone who didn't/couldn't join."*

4. Using virtual field experiences in combination with on-location field trips
*"Using Google Earth to conduct virtual field trips was difficult and not the same as*
*an in-person field trip but combining the use of Google Earth with in-person trips*
*may be beneficial."*

*"I think some of the resources we used in online learning were extremely helpful,*
*such as the Google Earth stops and the images of the outcrops in better*
*conditions. I don't think they substitute for the in-person experience, but if field*
*trips might become a mix of in-person observation and data collection plus*
*recorded/online chalk talks, it might be beneficial."*

As the SST instructors transition back into an environment where on-location
field trips are once again possible (we hope!) the MAAOT virtual field experiences are
being used to augment the five on-location field trips. We envision that students will
benefit from the tactile, on-the-outcrop experience of on-location field trips, but will also
appreciate the added perspectives of the virtual field experiences to enhance the
learning and review process. For students that may be unable to visit certain outcrops,
the virtual field experience will provide them with a way to investigate the outcrop and
participate with their group members in a meaningful and knowledgeable way.
Ultimately, the authors view this hybrid approach as a more inclusive approach to field-
based learning and a richer pedagogical experience for all students.




**6. Conclusions**
Virtual learning, whether in the classroom, lab, or in the field, may not be an appealing
or effective solution for all students. Interestingly, students that attended SST in-person
were more supportive of virtual learning options, perhaps reflecting a desire that these
options had been available when they took the course. A key consideration is that some
traditional on-location field experiences can be challenging for students with physical
and other disabilities, and geoscience departments need to have alternatives in order to
accommodate all current and prospective students. This is not only an ethical obligation,
but also important from a recruitment perspective, where geoscience educators need to
welcome students from all backgrounds in order to ensure the continued health of the
discipline.

Another consideration is the continuing uncertainty of the COVID pandemic
situation and the possible impacts of future variants. As the Fall 2021 semester begins,
we are witnessing another global uptick in COVID cases, underscoring the possibility of
a return to travel and field access restrictions at some point in the future. With the
development of virtual field experiences, such as those included in the MAAOT project,
instructors have alternative options if on-location access to field sites is restricted. The
necessity for virtual field options has always existed for some geoscience students, but
the COVID pandemic has made all of us realize that these virtual options need to be
available to the full community of students and instructors.

**Author Contributions**
All authors contributed to the writing of the manuscript. HL drafted and administered the
student survey and collected the student data.

**Competing interests**
The authors declare that they have no conflicts of interest.

**Acknowledgements**
The authors want to thank all of the SST students over the years that have participated
in MAAOT field trips and provided their thoughts and perspectives on the project.
Particular thanks go to the 32 students that responded to our online survey.



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
