# Peer review of "Virtual Learning Questionnaire for SST: Stratigraphy, Structure, Tectonics"

_Solid Earth, 2021_

## Author Response (AR1)

**Responses to reviewers**

Comments from Referee #1 Terry Pavlis (CC1 & RC1)
This is a very well written, insightful paper outlining experiences in using a virtual field trip format to replace a long standing in person field class conducted at James Madison University, Virginia, USA. The authors clearly state the history and objectives of the virtual field trip development and provide a clear assessment of the results of running the trip. I really appreciated the pros and cons of conventional vs virtual methods, in part because I think they are spot on based on my experience, but also because it is important for these things to be pointed out. All in all, a really nice paper.

The paper is well enough written that it could be essentially published as it. I have a few minor knit picky points linked to line numbers below, but all of them are very minor. My only significant comment is I think a lot of figures could be improved. There were very few annotations on the figures that I could read, no doubt because most of the figures are screen shots from the virtual field trip. For some things this is trivial, but for some of the images where there are important annotations that the students would see, this makes it hard to grasp what is presented to the students. This is an easy fix; if you use something like illustrator you could just reproduce the text box over the screen shot. You might be able to do more high resolution screenshots too? Depends on the software I would guess.

Specific comments keyed to text:
98-100: maybe clarify that these represent clastic wedges shed from the orogenic highland?
*Clarified in the text.*
101: reference figure 1
*Done.*
115: drop the "will"—future tense not necessary here
*Fixed.*
291: Might be worth mentioning here that 3D terrain models can allow virtual measurements quite easily, so had those been included this could have been done. There are several sources for this, but my former student's paper (Brush et al., 2018, Geosphere) has a description of using the method. There are a lot of other references, most cited in that paper; or you could reference the QGIS extension qsurf, which is an open source solution to the problem.
*The possibilities of 3D terrain models for virtual measurements are now mentioned and cited in the text.*
312: I wouldn't use the term faulting here; that means brittle deformation to most people. I'd say ductile shear or ductile shear zone development
*Clarified in the text as "a ductile fault (shear zone)".*
332: This whole section suffers from lack of clarity on how much of the field trip mechanics is automated and how much relies on direct communication with the instructor. This is an important point for anyone who wants to export what you've learned here to their exact situation.

*We have reorganized and rewritten Section 3 to clarify the field trip mechanics and how stratigraphic analyses are conducted on virtual field trips, in comparison to how this is done on traditional on-location field trips.*

430: "lousy weather" may not translate well in this European journal. Bad weather would be better!

*Changed to "bad weather"*

498: say "virtual class"; as worded it could be read as they "virtually indicated" as in they nearly all indicated.

*Changed to "virtual class"*

677: You might note that it might be more than tactile. I'm not an expert on memory, but I have certainly read popular science articles that indicate that environmental factors can be powerful long term memory influences. Something as simple as smells or sounds could impact people's ability to remember things more than 5 minutes at the outcrop. Might be worth a comment?

*We added a sentence to address this point.*

Most of the figures with Google earth screen shots have illegible lettering. Not sure how to fix this, but it is a problem. Can you use a different monitor for the screen shot and reduce the figure to make higher resolution images?

*We have upgraded the figures in higher resolution for easy readability*

Links are not live on the pdf I have, so I couldn't easily jump the NAGT website referenced

*Links should be live in the online version of the manuscript.*

Comments from Anonymous Referee #2 (RC2)

The first three sections focus on providing a detailed descript of the course itself. Traditionally, in APA formatting a lot of this content might be reserved for a "present study" section or within the Methods section. Related, a lot of the background for the survey design on historical inclusion of diversity within geoscience could be moved into an "Introduction" section. Please check on the requirements of the journal (this might be okay and just an organizational style I am not used to). It does read clearly, but feels more conversational and less theoretically-driven.

*We have substantially rewritten and reorganized the introductory sections of the manuscript to clarify the purpose and focus of this work. Hopefully, it is now apparent that this research primarily focused on a comparison of virtual and on-location field experiences for students, and less on theories of student learning.*

Related, each paragraph seems to begin by identifying the personal experience of the educator, how they went about trying to make the virtual field trip the same as the actual field trip, and then (sometimes) reference to previous literature. This likely reflects the experience of the educator, but I wonder if the theoretical argument of the manuscript would be strengthened by prioritizing theory that then drives the development of the curriculum? For example, there are no citation provided in section 3.3, suggesting everything discussed was from the educators' experience.

*We have reorganized and rewritten Section 3 to put our experiences and approaches in the context of other existing work on geoscience field educational approaches. Redundancies have been removed, and several pertinent references to existing work have been incorporated.*

Another example is the paragraph beginning on line 247, which begins by identifying the need to adjust teaching styles. There is a body of research examining teaching styles in online versus in-person contexts – why not begin with relevant theoretically-motivated, empirically-evaluated principles and then show how they were used to design the curriculum? In this way, the sections of the introduction might be framed around key issues/concerns when developing any field experience in a virtual space.

*We have rewritten and reorganized the "3. Experiences with Virtual Field Trips" sections, adding a new introductory paragraph to the section and reorganizing the "3.1 Instructor Experiences with Virtual Field Trips" section to clarify the background principles that governed our approaches to designing and conducting virtual field experiences. Background literature on transitioning to virtual education settings and virtual field environments has been incorporated and cited.*

In addition, I think these introductory sections would be strengthened by adding a focus from cognitive psychology and education on people's reasoning within real versus virtual environments. For example, are people able to navigate in the same way in both spaces? Can people develop mental maps of space when navigating through virtually vs. physically?

*We agree that the participation of a cognitive psychologist in this study would have strengthened the evaluation of student experiences in virtual and on-location field experiences. Unfortunately, we did not have the foresight to have a cognitive psychologist involved in this effort before and during the COVID pandemic. Perhaps we can address this component in related future work.*

The section on Community Access to Virtual Field Experiences does not seem to be part of a larger logical argument. How does this section inform the current study? Did the current study use these materials? Take similar methodological approaches? (I think this may be the aim of the section, but should be explicitly identified).

*We moved the information about the NAGT community access portal to the introduction, so that we could put our work into the general context of community efforts. This has streamlined the focus of the manuscript.*

Could you provide more information regarding the methods?

- What questions were asked? Or if too many to list all, could you provide the number of questions under each theme and a metric for internal reliability? Were they open-ended questions or multiple choice?

*The questions in the questionnaire are included as a supplement to the manuscript, although this supplement may have been omitted in the original manuscript files - sorry. We will make sure it is included in the revision files.*

- How was the data were coded into themes? Did you obtain a metric of inter-coder reliability?

*Data from the questionnaire were either numerical scores for student self-assessment or open-ended questions to encourage reflective comments from students. Coding of open-ended questions was straightforward classification into the themes: "Disability Access and Inclusion in the field", "Perceptions of virtual field experiences in comparison with on-locations field trips", "Incorporating mobile technologies". Coding was done by one of the authors.*

- Were the respondents' demographic information similar across cohorts? What span of years were the previous students drawn from (e.g., the last 1, 5, 10 years)? Is the course

offered in the Spring and Fall semesters (in which case previous students in the Spring semester may have different experiences then offerings in the fall because they have more student experience)? Did the same educator teach in the previous years (if different this may have influenced the results)?

*Demographic information was not collected on the questionnaire, which is now stated in the text. The questionnaire was sent to students from the recent virtual course as well as from the previous 5 years of the course. The SST course is a Fall semester course taught by the same instructors each year. All of this is now indicated in the text.*

The introduction should discuss preferred learning styles (which is a technical term in education) since that is a main outcome of the survey. To note, there is research showing that aligning teaching styles with students preferred learning styles does \*not\* improve learning. That student learning is more influenced by aligning appropriate teaching styles as per the content demands

*Student learning styles are not the focus of this manuscript, and we recognize that the focus of the manuscript (comparing virtual field experiences with on-location field experiences) was not sufficiently clear. Thus, we have modified the Introduction to more explicitly articulate the purpose and focus of the manuscript. Later in the manuscript we make sure to reference literature on student learning in field settings and put our experiences within that context.*

*Note that we added a final section to the Discussion: "5.3 Future Impacts of Virtual Field Experiences." Since we were able to return to on-location field trips for the Fall 2021 SST class, we are able to provide some insights into how our experiences with virtual field trips in Fall 2020 have impacted and changed how we conduct on-location field trips. When we wrote the original version of the manuscript we weren't in a position to comment on potential changes, but this is now an informative addition to the discussions.*

---

## Author Response (AR2)

**Response to Topical Editor**

Comments from Topical Editor Marlene Villeneuve
Can you please address the following:

From: The GC editorial "Geoscience Communication – Building bridges, not walls"(Section 4 point 2):

"All research articles should include an explicitly marked section that considers the ethics of the investigation and should also demonstrate how the research has received ethical clearance from their research institute or professional body. If this is not possible, then a clear rationale should be given for any extenuating circumstances, ideally in the cover letter to the editor upon submission of the manuscript. Furthermore, if institutional ethical approval is not possible (e.g. if you are an independent researcher), then the ethical guidelines for a country or governing body should be adhered to: for example, the British Educational Research Association (BERA) provides ethical guidelines for educational research (see e.g. Flewitt, 2005)."

https://gc.copernicus.org/articles/1/1/2018/

*We appreciate the comments from the topical editor to clarify our ethical treatment of the survey data collection and reporting. We note that this manuscript was submitted to Solid Earth, not to Geoscience Communication. However, as the manuscript was submitted to a SE/GC joint special issue, we agree that it is important to make it clear to readers how the survey data was handled. Thus, we have rewritten the third paragraph of section 4, lines 490-505. We have made it clear that no identifying information was collected from the survey respondents, and no demographic data was collected. In addition, survey data was aggregated for analysis and no identifying data was preserved or added in the aggregation process. All survey respondents had the option to "opt out" of answering any question or part of the survey, and also could indicate if they didn't want their survey responses included in the data used for publication. This methodology is all in accordance with the ethical requirements for collecting survey data from students as defined by the James Madison University Institutional Review Board (IRB.) As required by the IRB, we had our survey data collection instrument and protocol for analysis and reporting verified and approved by the IRB (Protocol number 21-2232.)*